# OpenReview forum: "Beyond ordinary Lipschitz constraints: Differentially Private optimization with TNC"
_TMLR — Accepted by TMLR_

### Review · Reviewer_K3oy · 2025-06-03

**Summary Of Contributions:**

This paper addresses differentially private stochastic convex optimization (DP-SCO) under the Tsybakov Noise Condition (TNC), targeting settings where the loss function may not be Lipschitz or where the Lipschitz constant is very large. The authors propose novel algorithms that achieve high-probability excess population risk bounds based only on the boundedness of higher-order gradient moments and the TNC, rather than Lipschitz continuity. They further present tight lower bounds under $\rho$-zCDP, demonstrating that their results are near-optimal.

**Audience:**

Yes

**Broader Impact Concerns:**

No concerns. This is a theoretical work that may have societal impacts in the future.

**Claims And Evidence:**

Yes

**Requested Changes:**

- See weaknesses.
- Ensure consistent and appropriate usage of LaTeX citation commands such as \cite, \citep, and \citet. For example, citation formatting needs to be reviewed and possibly corrected in the following parts:
  - Caption of Definition 2 and 4
  - The paragraph following Definition 7
  - Remark 2
  - The paragraph below Theorem 6

**Strengths And Weaknesses:**

Strengths:

- By leveraging the TNC and assuming heavy-tailed gradients, the paper relaxes commonly used assumptions (e.g., bounded Lipschitzness, sub-Gaussian input distributions), thereby broadening the applicability of DP-SCO to more realistic settings.

- The proposed algorithms yield utility bounds that are independent of the Lipschitz constant—an important advancement, especially in scenarios involving heavy-tailed data. The combination of high-probability upper bounds and tight lower bounds further strengthens the theoretical contribution.

- The paper is well-written and organized. Despite the technical depth, the exposition is clear, definitions and assumptions are well-motivated, and comparisons to prior work are comprehensive and informative.



Weaknesses:

- While the adoption of the TNC leads to significant theoretical improvements, its practical relevance could be more clearly articulated. The paper cites past papers and gives examples such as $\ell_1$-constrained $\ell_4$-norm linear regression, but these may not be sufficiently accessible or persuasive for readers unfamiliar with TNC. I encourage the authors to include additional concrete examples or offer stronger justification for the prevalence and importance of TNC in real-world applications. Moreover, it would be helpful to explain under what circumstances the TNC condition can be verified and how the parameter $\theta$ (or $\overline{\theta}$) might be estimated in practice.

- The algorithms depend on several hyperparameters (e.g., $C_i$, $\lambda_i$) that are tied to abstract quantities like the $k$-th moment of gradients or the TNC parameter. These parameters may be difficult to estimate or tune in practice, especially when the required quantities are unknown.

- The paper does not provide any numerical results. While this is not a strict requirement for a theoretical contribution, some simulation studies — particularly comparing standard optimization methods to the proposed ones under TNC—would significantly enhance the paper. Such experiments would not only illustrate the practical impact of the TNC assumption but also make the theory more accessible to a broader audience.

---

> ### Author Response · Authors · 2025-07-15
>
> >**W1**: More examples on TNC.
>
> **Response:** More examples are shown in [1].
>
>
> >**W2**: Hyperparameters are unkown and hard to estimate.
>
> **Response:** By following the expression of the parameters shown in the algorithm you can calculate them directly. As for the common parameter settings, I will clarify in the numerical experiments in  response to Weakness 3.
>
>
>
> >**W3**: Numerical experiments.
>
> **Response:** We implement some numerical experiments on a8a, a9a and  w7a with our algorithm 3 and DP SGD. Though DP SGD performs better than ours, we need to clarify that the condition and requirements of our algorithm and DP SGD are different. We finish two groups of exprements. The first one is on the varing sample size, ranging from 5000 to 30000, with fixed privacy budget $\varepsilon=8$. In the second one we alter the privacy budget $\varepsilon$ in the following list: [0.5, 1.0, 1.5, 2.0, 3.0, 4.0, 5.0] with fixed sample size $n=10000$. Besides, we set $\delta =\frac{1}{n^{1.1}}$
>
> a8a:
> | ε   | LNC-GM Test MSE | DP-SGD Test MSE |
> | --- | --------------- | -------------------- |
> | 0.5 | 0.7072          | 0.6115               |
> | 1.0 | 0.6380          | 0.4958               |
> | 1.5 | 0.6131          | 0.4765               |
> | 2.0 | 0.6082          | 0.4703               |
> | 3.0 | 0.6041          | 0.4661               |
> | 4.0 | 0.6024          | 0.4647               |
> | 5.0 | 0.6014          | 0.4641               |
>
> | n      | LNC-GM Test MSE | DP-SGD Test MSE |
> | ------ | --------------- | -------------------- |
> | 2 000  | 0.6687          | 0.5165               |
> | 5 000  | 0.6380          | 0.4861               |
> | 10 000 | 0.5850          | 0.4911               |
> | 15 000 | 0.5665          | 0.4803               |
> | 20 000 | 0.5487          | 0.4684               |
> | 22 696 | 0.5584          | 0.4811               |
>
>
> a9a:
> | ε   | LNC-GM Test MSE | DP-SGD Test MSE |
> | --- | --------------- | -------------------- |
> | 0.5 | 0.7017          | 0.6163               |
> | 1.0 | 0.6346          | 0.5099               |
> | 1.5 | 0.6089          | 0.4917               |
> | 2.0 | 0.6042          | 0.4856               |
> | 3.0 | 0.6003          | 0.4815               |
> | 4.0 | 0.5988          | 0.4802               |
> | 5.0 | 0.5979          | 0.4797               |
>
> | n      | LNC-GM Test MSE | DP-SGD Test MSE |
> | ------ | --------------- | -------------------- |
> | 2 000  | 0.6737          | 0.5212               |
> | 5 000  | 0.6346          | 0.4877               |
> | 10 000 | 0.5867          | 0.4921               |
> | 15 000 | 0.5691          | 0.4827               |
> | 20 000 | 0.5516          | 0.4722               |
> | 30 000 | 0.5457          | 0.4809               |
>
> w7a:
> | ε   | LNC-GM Test MSE | DP-SGD Test MSE |
> | --- | --------------- | -------------------- |
> | 0.5 | 0.1161          | 0.2357               |
> | 1.0 | 0.1139          | 0.1388               |
> | 1.5 | 0.1122          | 0.1242               |
> | 2.0 | 0.1108          | 0.1192               |
> | 3.0 | 0.1099          | 0.1149               |
> | 4.0 | 0.1100          | 0.1127               |
> | 5.0 | 0.1102          | 0.1111               |
>
> | n      | LNC-GM Test MSE | DP-SGD Test MSE |
> | ------ | --------------- | -------------------- |
> | 2 000  | 0.1161          | 0.1135               |
> | 5 000  | 0.1139          | 0.1154               |
> | 10 000 | 0.1168          | 0.1078               |
> | 15 000 | 0.1112          | 0.1063               |
> | 20 000 | 0.1122          | 0.1077               |
> | 24 692 | 0.1099          | 0.1022               |
>
> >**Requested Changes**:
>
> **Response:** Thanks for pointing this out. I will correct them in the next version.
> [1] Jinyan Su, Lijie Hu, Di Wang. Faster Rates of Differentially Private Stochastic Convex Optimization.

---

> ### Author Response · Authors · 2025-07-21
>
> **W1**: More examples on TNC.
>
> **Response:** Please check the example in red on Page 8 of the revised version.
>
> **W2**: Explain under what circumstances the TNC condition can be verified and how the parameter $\theta$ (or $\overline{\theta}$) might be estimated in practice.
>
> **Response:** I have clarified in red text on Page 5 of the revised version.
>
> >**W3**: Hyperparameters are unknown and hard to estimate.
>
> **Response:** By following the expression of the parameters shown in the algorithm, you can calculate them directly.  As for the common parameter settings, I will clarify in the numerical experiments in response to Weakness 4.
>
>
> >**W4**: Numerical experiments.
>
> **Response:** I have added the numerical experiments in Section 7 on Page 11 of the revised version. And the figures are shown in Appendix A due to the limited space.
>
> >**Requested Changes**:
>
> **Response:** I will correct them in the revised version.

---

### Review · Reviewer_ZoeR · 2025-06-09

**Summary Of Contributions:**

This paper studies the bound of differentially private optimization. While this problem has already been widely studied in many previous literatures, This paper introduces a Tsybakov noise condition. In particular, this paper assumes that

$$F(w)-F(w^)\geq \lambda ||\theta-\theta^||_2^\theta$$

When $\theta = 0$, the condition reduces to general convex case. The bound derived in this paper also matches with existing works.

When $\theta=2$, the condition becomes the strong convex case. The bound derived in this paper matches existing works about strong convex case.

**Audience:**

Yes

**Broader Impact Concerns:**

I do not have any ethical concerns.

**Claims And Evidence:**

Yes

**Requested Changes:**

I do not have any requested changes.

**Strengths And Weaknesses:**

In general, I think that this paper indeed provides a great analysis of DP optimization problem with heavy-tailed response. Based on my reading, this paper makes a solid contribution. Moreover, I have briefly read through the proof. It seems that the result is technically correct. Therefore, I strongly recommend acceptance at TMLR.

---

> ### Author Response · Authors · 2025-07-15
>
> Thank you for taking the time and effort to review this manuscript.

---

### Review · Reviewer_sWaU · 2025-07-08

**Summary Of Contributions:**

This submission considers DP stochastic convex optimization. Most work in DP-SCO relies on strong assumptions about the loss being uniformly Lipschitz; this work lies in a "heavy tailed" setting where we assume a bound on the low-order moments of the gradients. One of the specific assumptions used is that of a "Tsybakov Noise Condition," an assumption from the optimization and machine learning literature that might allow improved results for important special cases.

**Audience:**

Yes

**Broader Impact Concerns:**

None.

**Claims And Evidence:**

Yes

**Requested Changes:**

Can you restate Theorem 1 in a way that makes the role of $p,\delta,$ and $\beta$ precise?

Can you provide an overview of which of your theorems improve over those of Lowy and Razavivyan, with pointers to their theorem numbers?

Can you briefly compare to the 2024 work of Asi, Liu, and Tian?

**Strengths And Weaknesses:**

I am not intimately familiar with prior work, but it seems like the analysis overcomes several technical challenges. The work will surely be of interest to the TMLR audience.

I found some of the technical presentation and discussion confusing. Here are some questions I had about Theorem 1.
- Why is ``$p$'' in the theorem? What purpose does it serve? It doesn't show up in the conclusion.
- Where is the privacy parameter $\delta$ in the conclusion?
- Why do we restrict the failure probability $\beta$ to be at most $1/n$? This doesn't make sense to me.
- Where is $\beta$ in the conclusion? It seems to be hidden as a logarithmic factor, but since the discussion of high-probability versus in-expectation is important here the dependence on $\beta$ should be explicit.

I also have some questions about the relationship with prior work. We seem to be missing a reference to [1]. I'm also confused about the formal relationships with the work of Lowy and Razavivyan; part of their contribution is to obtain results without the smoothness assumptions used here, so it's not clear to me which results are directly comparable.

The current application of citations throughout the text is hard to read and inconsistent.

[1] Asi, H., Liu, D., & Tian, K. Private stochastic convex optimization with heavy tails: Near-optimality from simple reductions. NeurIPS 2024.

---

> ### Author Response · Authors · 2025-07-15
>
> >**W1**: Why is there no dependence on p in the final bound?
>
> **Response:** p is used to control the Lipschitz constant thus we can remove the Lipschitz constant from the final bound. We can see that in the proof of Thoerem 10, there exists a term with $n^p$ both in the numerators and denominators. By assuming that $L_f$ is controlled by the $O(n^p/2)$ and choosing specific $\eta$, we can eliminate the p in the final bound. A similar result holds for Theorem 2.
>
> >**W2**: Why is there no dependence on $\delta$ in the final bound?
>
> **Response:** Thanks for pointing this out. Actually, I omitted the absolute constants in the state of the theorems. $\delta$ should be with the dimension $d$. In theorem 1, the formal result should be $O( \tilde{R}_{2k,n}D( \frac{1}{\sqrt{n}}+(\frac{\sqrt{d \log(1/\delta) \log n}}{\epsilon  n})^{\frac{k-1}{k}})$ $+\frac{D\sqrt{\log (1/\beta )}}{2^{p+1}\sqrt[]{n}})$. I will formalize it in the next version.
>
> >**W3**: Why  $\beta$ should be less than $\frac{1}{n}$?
>
> **Response:** It is the requirement of the Lemma, which is Theorem 7 in [1], which is shown in the proof, Lemma 3 on Page 13.
>
> >**W4**: Why is there no dependence on $\beta$ in the final bound?
>
> **Response:** I omit the $\beta$, which is an absolute constant, in the statement of Theorem 1. You can check the full expression in the response to Weakness 2.
>
> >**Requested Changes 1**: Restatement of Theorem 1.
>
> **Response:** See the response to weakness 2.
>
> >**Requested Changes 2**: Can you provide an overview of which of your theorems improve over those of Lowy and Razavivyan, with pointers to their theorem numbers?
>
> **Response:** Firstly, we demonstrate the high probability bound instead of the expectation form compared the Theorem 6 in [2].  We introduce　TNC into the following theorems in our paper and generalize the assumptions, which have nothing to do with the theorems in [2].
>
> >**Requested Changes 3**: Can you briefly compare to the 2024 work of Asi, Liu, and Tian?
>
> **Response:** We use different algorithms from theirs, and our assumptions are distinct from theirs though our algorithms can both deal with heavy tailed data. In addition, we further derive the bound for functions satisfying TNC and remove the dependence on Lipschitzness of the loss functions.
>
> [1] Hilal Asi, Daniel Levy, John Duchi. Adapting to Function Difficulty and Growth Conditions in Private Optimization.
>
> [2] Lowy and Razavivyan, Private Stochastic Optimization with Large Worst-Case Lipschitz Parameter: Optimal Rates for (Non-Smooth) Convex Losses and Extension to Non-Convex Losses

---

> ### Author Response · Authors · 2025-07-21
>
> >**W1**: Why is there no dependence on p in the final bound?
>
> **Response:** I clarify in Remark 2. Please check the highlighted part in red. p is used to control the Lipschitz constant; thus, we can remove the Lipschitz constant from the final bound. We can see that in the proof of Theorem 10, there exists a term with $n^p$ both in the numerators and denominators. By assuming that $L_f$ is controlled by the $O(n^p/2)$ and choosing specific $\eta$, we can eliminate the p in the final bound. A similar result holds for Theorem 2.
>
> >**W2**: Why is there no dependence on $\delta$ in the final bound?
>
> **Response:** Thanks for pointing this out. I have revised the statement of the theorem to avoid omitting these constants. Please check highlighted part in each theorem.
>
> >**W3**: Why  $\beta$ should be less than $\frac{1}{n}$?
>
> **Response:** It is the requirement of the Lemma, which is Theorem 7 in [1], which is shown in the proof, Lemma 3 on Page 13.
>
> >**W4**: Why is there no dependence on $\beta$ in the final bound?
>
> **Response:** I have revised the statement of the theorem to avoid omitting these constants. Please check highlighted part in each theorem.
>
> >**Requested Changes 1**: Restatement of Theorem 1.
>
> **Response:** Please see the full statement of Theorem 1 on Page 6 in revised version.
>
> >**Requested Changes 2**: Can you provide an overview of which of your theorems improve over those of Lowy and Razavivyan, with pointers to their theorem numbers?
>
> **Response:** I clarify the difference in Related Works on Page 3 of revised version. Please check the parts in red.
>
> >**Requested Changes 3**: Can you briefly compare to the 2024 work of Asi, Liu, and Tian?
>
> **Response:** I make some comparisons in Related Works on Page 3 of revised version. Please check the parts in red.

---

> > ### Comment · Reviewer_sWaU · 2025-08-14
> > **Comparison with Theorem 6 of Lowy & Razavivyan**
> >
> > If I am reading them right:
> > - Theorem 6 of Lowy & Razavivyan does not assume that the loss function satisfies a smoothness condition, but
> > - Theorem 1 of this submission assumes that the loss function is smooth.
> > Thus it seems that Theorem 1 is not a strict improvement over their work.
> >
> > In Theorem 1, why can't I always take $p$ to be sufficiently large, removing it from the assumptions and the conclusion? I seem to be missing something.

---

> > > ### Author Response · Authors · 2025-08-14
> > >
> > > Thanks for replying to my comments.
> > >
> > > >**Q1**: The requirement of smoothness for the loss functions.
> > >
> > > **Response:** Our paper and theirs focus on different questions, which cannot be regarded as simple improvement. Actually, $\alpha$-smoothness is necessary for the proof of Theorem 2 instead of Theorem 1. We can remove this condition from Theorem 1 in next version. The reason why I add this in the manuscipt is to keep consistence with Theorem 2.
> > >
> > > >**Q2**: The role of $p$ in Theorem 1.
> > >
> > > **Response:** I clarified the role of $p$ in the revised version, where I show the complete version of the bound with $p$. You can check the final part of the proof of Theorem 1 on Page 21. If $p$ is sufficiently large, then we need to pick tiny $\eta$ and adjust other corresponding parameters such as $T, D$ etc. , leading to bad performance in practice. In addition, $p$ cannot be removed as we need the Lipschitzness of the loss functions for the lemma used in the proof of Theorem 1.

---

> > > > ### Author Response · Authors · 2025-08-14
> > > >
> > > > >**Q1**: Further explanation of the difference between the two papers and the role of smoothness.
> > > >
> > > > **Response:**Our paper and theirs focus on different problems, which cannot be regarded as simple improvement. We reach the high probability bound and they derive the bound in expectation. Actually, $\alpha$-smoothness is necessary for the proof of Theorem 2 instead of Theorem 1. We can remove this condition from Theorem 1 because the restriction of the stepsize $\eta$ is critical in the proof for the TNC case only.
> > > >
> > > > >**Q2**: Further explanation of the role of $p$.
> > > >
> > > > **Response:** $p$ can be indeed sufficiently large, causing it to disappear from the final bound. However, in accordance with reviewer sWaU’s request, and to clarify the role of $p$ for the reader, we present the complete form in Theorem 1.

---

### Comment · Reviewer_ZoeR · 2025-06-01

This paper studies the bound of differentially private optimization. While this problem has already been widely studied in many previous literatures, This paper introduces a Tsybakov noise condition. In particular, this paper assumes that

$$F(w)-F(w^*)\geq \lambda ||\theta-\theta^*||_2^\theta$$

When $\theta = 0$, the condition reduces to general convex case. The bound derived in this paper also matches with existing works.

When $\theta = 2$, the condition becomes the strong convex case. The bound derived in this paper matches existing works about strong convex case.

In general, I think that this paper indeed provides a great analysis of DP optimization problem with heavy-tailed response. Based on my reading, this paper makes a solid contribution. Moreover, I have briefly read through the proof. It seems that the result is technically correct. Therefore, I strongly recommend acceptance at TMLR.

---

### Decision · Action_Editor_1JVZ · 2025-08-23

**Recommendation:** Accept as is

**Additional Comments:**

There are two times "Accept" and one "Leaning Accept" so this is a clear accept. This seems to be a strong paper, however I remain reluctant to tag the "featured certification" due to the following reason (also, I think the authors have had the chance to clarify these issues based on the review and comments of Reviewer sWaU).

As pointed out by Reviewer sWaU, some confusion remains regarding how exactly is the paper related to the prior works by Asi et al. (2024) and Lowy and Razaviyayn (2023). In the first version of the work, the reference to Asi et al. (2024) was omitted. It is still unclear which results are more generally applicable. This paper requires smoothness of the loss function (see the assumptions of Thm. 1) while, e.g., the analysis of Lowy and Razaviyayn (2023) (see their [Thm. 6](https://proceedings.mlr.press/v201/lowy23a/lowy23a.pdf)) does not make smoothness assumptions. The added material does fully clarify the differences.

I would still recommend addressing for the final version the comments by reviewer sWaU regarding how to put the results into context, meaning, e.g., how to relate the resutls to the those of the prior works by Asi et al. (2024) and Lowy and Razaviyayn (2023).

Please also polish the newly added material. Examples:

- The phrasing in this sentence is slightly cumbersome : "In contrast to the work of (Lowy & Razaviyayn, 2023), our approach builds upon Theorem 6 from (Lowy & Razaviyayn, 2023)."
- "Furthermore, our analysis extends toTNC"
- Remark 2: write log with $\log$
- bound of Thm. 2: one extra $+$
- page 8: x -> $x$

**Audience:**

Yes

**Audience Explanation:**

Based on the reviews, all the reviewers seem to think that the paper fits well to TMLR. The paper deals with a fundamental problem in DP machine learning (DP-ERM) so I also think it definitely fits well for TMLR.

**Claims And Evidence:**

Yes

**Claims Explanation:**

Based on the reading of the paper and the reviews, the analysis seems solid and the bounds correct.